# Evaluation of Colombian Crops Fibrous Byproducts for Potential Applications in Sustainable Building Acoustics

**DOI:** 10.3390/polym13010101

**Published:** 2020-12-29

**Authors:** Tomas Simon Gomez, Santiago Zuluaga, Maritza Jimenez, María de los Ángeles Navacerrada, María del Mar Barbero-Barrera, Daniel de la Prida, Adriana Restrepo-Osorio, Patricia Fernández-Morales

**Affiliations:** 1Escuela de Ingenierías, Universidad Pontificia Bolivariana, Circular 1 # 70-01, Medellín 050031, Colombia; tomas.s.gomez.m@gmail.com (T.S.G.); santiago.zuluaga@upb.edu.co (S.Z.); maritza.jimenez@upb.edu.co (M.J.); adriana.restrepo@upb.edu.co (A.R.-O.); patricia.fernandez@upb.edu.co (P.F.-M.); 2Escuela Técnica Superior de Arquitectura, Universidad Politécnica de Madrid, Avenida Juan de Herrera 4, 28040 Madrid, Spain; mar.barbero@upm.es (M.d.M.B.-B.); d.delaprida@alumnos.upm.es (D.d.l.P.)

**Keywords:** fique, coir, sustainable materials, sound absorption measurement, natural fibers

## Abstract

Local production of construction materials is a valuable tool for improving the building sector sustainability. In this sense, the use of lignocellulosic fibers from local species becomes an interesting alternative to the development of such materials. As it is thought that the properties of fiber-based materials are dependent on the fibers properties, the knowledge of such properties is fundamental to promote materials development. This study compares the physical, morphological, acoustic, and mechanical characteristics of coir (Cocos nucifera) and fique (Furcraea Agavaceae) fibers and panels. The chemical composition appears to be associated with the general behavior of the fibers and panels, regarding higher tensile strength, thermal degradation behavior, and water absorption. In most tests, fique had the upper hand, showing superior performance; however, on thermal degradation and water absorption, both materials had similar behavior. The sound absorption measurement showed that the fiber diameter affects the sound absorption at high frequencies, where fique panels showed better performance than coir panels.

## 1. Introduction

International trends that seek to promote the achievement of the Sustainable Development Goals (SDG) related to sustainable cities and communities, and responsible consumption and production [1], have favored research and development about new applications or techniques that facilitate the transition to sustainable economies. In particular, for the building economic sector, the interest in new alternative materials with low environmental impact and energy efficiency has grown. Materials based on natural fibers are becoming increasingly popular, due to some common features such as their high availability and environmental sustainability benefits [2,3].

From an availability perspective, it should be noted that, globally, over 64 million tons of natural fibers are being consumed each year, mostly by the textile industry [4]. Therefore, with a considerable amount of fibrous wastes, new strategies have been created to develop materials, using biodegradable or recycled elements such as discarded natural fibers [5]. Considerations about natural fiber environmental benefits include the reuse of waste and also renewability and energy efficiency, contributing to a reduction of the environmental impact of traditional materials in a variety of engineering applications [6].

Moreover, several properties of natural fibers have proven to be an alternative to synthetic fibers in a variety of applications [6]; among them, low mass density and cell structure make them potential materials for indoor acoustic and thermal conditioning [3], which, in some cases, have outperformed synthetic fibers [7]. Examples of common natural materials used for sound damping in real scenarios include those with high chemical stability and high fire resistance, such as cork, reed, and green walls [8,9,10]. Despite the natural fiber advantages, there are still few commercial applications, mainly due to their disadvantages like a limited fungal, parasite, and fire resistance, which require the application of chemical treatments that reduce their sustainability ratings [10], increase costs and also cause the perception of insecurity among users. Furthermore, the potential applications for materials derived from fibrous waste are highly dependent on the characteristics of the fiber, and this is the reason why it is vital to make an in-depth study of fibers properties.

The purpose of this research is to contribute to extending the knowledge of coir and fique fibers behavior. The main idea is to characterize these fibers and to compare their potential use as sound absorptive materials using natural fibers and fibrous waste of local origin as raw input for its development, which could generate a positive impact on local economies, waste management strategies and a huge contribution to their potential use in the building sector.

## 2. Natural Fibers Feature for Acoustic Applications

Coir and fique are natural fibers available in the equatorial zone. Coir fibers come from the mesocarp of the coconut fruit and represent around 30% of the net fruit weight, which is usually discarded or burnt [11]. On the other hand, fique fibers come from the fique plant leaves; where, the fique production is focused mainly on long fibers, leaving an estimated fibrous waste accounting for 35% of the net weight of the leaf [12]. Therefore, there could be high availability of these raw materials, which could be converted to various applications, such as, thermal, and acoustic insulation for facades and ceilings.

Coir acoustic properties have been studied more frequently, mainly in countries with large coconut industries such as Brazil, India, and Malaysia. Several authors have studied the acoustic properties of materials made of coir making use of different analytical models [13,14,15,16,17,18]. In these studies, the average diameter, weight, length, and volume of the fiber were considered for the sound absorption calculation. Likewise, the sound absorption coefficient of different natural materials, among them raw coir fibers, has been measured and modeled [6,7,19,20]. In contrast, fique-based materials acoustic applications have not been as profoundly studied as coir. Fique fibers have been used to manufacture nonwovens and tested for sound absorption with the impedance tube method; however, tested samples lack dissipation capabilities in the low-frequency range, where commercial products outclass them [21,22,23]. 

The acoustic properties of vegetable fibers extracted from leaves, husks, or fruits and stems have been studied [10,24,25,26,27]. It has been noted that many of these studies have shown satisfactory results where the acoustic properties were comparable to those of synthetic fibers [7,28]. However, the main properties of the fibers are not deeply studied for acoustic-oriented applications, nor are compared among different plant species.

In this study, characterization and comparison of fique and coir fiber in morphology, chemical composition, physical properties, and tensile strength were made. Additionally, samples of panels were fabricated using both fibers to evaluate water absorption and swelling properties, flexural strength, and morphology. Finally, the acoustic absorption of the materials was measured using an in-situ method.

## 3. Materials and Methods

Coir fibers (Cocotech Medellín, Colombia) and fique fibers (Cadefique, Guarne Colombia) were acquired from local growers in Antioquia (Colombia). The fibers were extracted by mechanical methods, cleaned, and dried by the suppliers.

In Section 3.1 and Section 3.2, the methods, equipment, and techniques used to characterize fiber properties and fiber panels properties are shown, respectively. It should be noted that hypothesis testing was implemented to compare obtained results among both fibers to determine if the means were significantly different; therefore, T-tests were used.

### 3.1. Fibers Characterization

#### 3.1.1. Chemical Composition

Coir and fique fibers cellulose, hemicellulose, lignin, and ash contents were quantified following standard ASTM D1106 and acid/neutral detergent fiber (ADF) (NDF) dissolution [29]. Additionally, to determine the dry weight required for the mentioned test, moisture content was measured using a moisture analyzer Mettler Toledo HE73 (Mettler Toledo, Colombus, OH, USA).

#### 3.1.2. Thermal Degradation

The thermal stability of the fibers was evaluated with thermogravimetric analysis (TGA), using Mettler Toledo TGA/SDTA85I° equipment (Mettler Toledo, Colombus, OH, USA). The weight loss (TG) and the derivative (DTG) curves were obtained under a nitrogen atmosphere with a flow rate of 40 mL/min. The experiments were run from 30 °C to 800 °C, at a heating rate of 10 °C/min.

#### 3.1.3. Microstructural Observation

The cross-section morphology of fique and coir fibers was observed using a scanning electron microscope (SEM) JEOL/OL JSM-6490 (JEOL Ltd., Akishima, Tokyo, Japan) operating at 20 kV. The fibers were lyophilized and fixed in a wax sample carrier, the cross-sections were exposed without damaging the internal structure by using a microtome, finally, samples were metalized with gold.

#### 3.1.4. Diameter and Density Measurement

The mean diameter size of individual fibers was obtained using the software ImageJ as an image analysis tool [30]. Images of the longitudinal section of the fibers were captured with a Motic stereomicroscope SMZ-143 FBGG (Motic, Hong Kong). The diameter was obtained using the super-pixel approximation, the diameter size in pixels is calculated and then converted into the units of interest from the scale in the images [31]. Additionally, the real density of the fibers was obtained by the gas displacement method, An AccuPyc II 1340 (Micrometrics, Ottawa, ON, Canada) helium gas pycnometer was used, 180 purges for each fiber type were made.

#### 3.1.5. Mechanical Properties

Mechanical properties were evaluated according to the standard ASTM D3822 based on the single filament test. The test was conducted with an Instron 5582 (Instron, Norwood, Massachusetts, USA) universal testing machine at a crosshead speed of 5 mm/min, a 50 N load cell, and a gauge length of 50 mm, at room temperature conditions, 40 fibers were tested for each type.

### 3.2. Panel Characterization

#### 3.2.1. Panels Manufacture

Coir and fique panels were manufactured using fibers with a mean length of 50 mm to which a natural rubber latex solution (Protokimica, Medellín, Colombia) was applied by spray method to avoid fibrous network saturation. Fibers were placed and mixed in a frame of 250 × 250 × 25 mm. The samples were compacted to the mold height and then set to dry in a convection oven at 70 °C for 2 h. For each fiber type, a set of 3 panels was fabricated.

#### 3.2.2. Microstructural Observations

Panel microstructure was observed using an optical digital microscope Jiusion F210 (Jiusion, China). Observations were made on the surface of the samples. Panels microstructure was observed before and after water immersion.

#### 3.2.3. Water Absorption and Thickness Swelling of Panels

Water absorption by total immersion (WA) was calculated according to the European Standard EN 12087. The water absorption of each panel was calculated following Equation (1). A total of 3 samples per panel type were submerged in a water tank at ambient temperature. Weight was recorded doubling time intervals starting at 5 min, and ending at 24 h when constant weight was reached, at which point the water was removed. The samples were set to dry at ambient temperature until a constant weight was reached, the drying rate was estimated from weight loss measurements made in analytical balance.
(1)$$Water\;absorption\;\left(\%\right)\;=\;\left(\frac{Msat-Md\;}{Md}\right)\;\times 100\%,$$
where *Msat* is the mass of the saturated panel in g, and *Md* is the dry mass of the panels in g.

Swelling after water immersion was measured according to the European Standard EN 317. Thickness swelling (TS) was calculated as a percentage of increase between before and after immersion of the samples in water for 24 h. A digital micrometer VOGEL DIN 862 (Vogel, Kevelaer, Germany) with a resolution of 0.01 mm was used, TS for each panel was determined following Equation (2).
(2)$$Thickness\;Swelling\;\left(\%\right)\;=\;\left(\frac{E1-E0}{E0}\right)\times 100\%,$$
where E0 and E1  are the thickness of the panels in cm, before and after immersion in water for 24 h, respectively.

#### 3.2.4. Mechanical Testing

A three-point flexural test was performed with a universal testing machine with a load cell of 5 kN. Modulus of rupture (MOR) was calculated with Equation (3) with the recorded tensile stress when displacement was 5% of the outer span. Since the span was 160 mm, the load for the flexural strength corresponded to 8 mm of displacement.
(3)$$MOR\;=\;\frac{3\;FL}{2be^{2}}\;,$$
where *MOR* is the modulus of rupture in MPa, *F* is the load in N, L, b, and e are the span between supports, the width, and thickness of the sample, respectively in mm.

#### 3.2.5. Sound Absorption Measurement

An impedance gun of Microflown Technologies (Microflown Technologies, Arnhem, Netherlands) was used to carry out the in-situ measurements of sound absorption. This impedance gun consists of a small spherical loudspeaker, and a pressure and velocity probe (PU Probe). The measuring frequency range is set between 200 Hz and 10 kHz. The lower limit is determined by the loudspeaker behavior, whose small dimensions do not allow the range to be extended to lower frequencies. On the other hand, the upper limit is established by the difficulty in calculating the phase of the impedance above this frequency, which could reduce the reliability of the results.

The loudspeaker emits a broadband noise, and the PU probe records both pressure and velocity signals. With this information, the absorption coefficients are calculated. This experiment was carried out using a Plane-wave model as it has been shown to provide equivalent to those obtained in an impedance tube [32].

## 4. Results and Discussion

### 4.1. Fiber Characterization Results

#### 4.1.1. Fiber Main Chemical Components

Cellulose is the basic structural component of all plant fibers, and it has been associated with a better mechanical performance of the fibers [33,34,35]. On the other hand, lignin has been associated with biodegradation resistance, and it is considered an encrusting agent in the cellulose/hemicellulose matrix and often is referred to as the plant cell wall adhesive [33,35,36]. The effects of the chemical composition on the properties of the fibers will be discussed in each section.

Table 1 shows the chemical composition of coir and fique fibers obtained by the ASTM D1106– 96 and the ADF-NDF methods. Fique fiber’s chemical composition coincides with literature reports with a major cellulose content over 50% and low lignin content [37,38,39]. In addition to that, cellulose (58.3%) and hemicellulose (14.6%) contents in fique fibers are higher than in coir. While coir fiber is mainly composed of cellulose (46.5%) and lignin (33.1%) [40,41,42,43,44].

Chemical content and diameter of natural fibers vary depending on the roles it plays in the plant. Since coir fibers come from the seed outer layer cellulose content is lower, while lignin and is expected to be considerably high as a deterrent for microbial attacks or predators [45], On the other hand, fique fibers come from the leaves, therefore, lignin content is less abundant, while fique is a branchless plant, its fibers are responsible for supporting the leaf weight, which based on its role, will probably show better mechanical performance [46].

#### 4.1.2. Thermal Behavior

TGA and DTG curves for coir are displayed in Figure 1. Coir thermogram showed 3 main thermal events. The first event occurs at temperatures lower than 100 °C and represented a 6% mass loss, which was attributed to the evaporation of moisture content, similar to reports by other studies [47,48,49]. The second thermal event started at 260 °C with a degradation peak at 275 °C, the recorded weight loss was close to 24%. There is a general agreement on the onset temperature for the degradation of hemicellulose and low molecular weight components, the mass loss recorded coincided with literature reports with values near 20% [47,48,49]. The third event was credited mainly to cellulose degradation, starting around 319 °C, with a weight loss of 25% and a degradation peak at 331 °C, literature reports on coir show an onset temperature slightly higher, close to 350–360 °C. Moreover, weight loss after degradation in coir agreed with our findings in a range from 25% to 30% [47,48,49].

The lignin degradation starts around 200 °C and peaked close to 430 °C following the trend of most natural fibers [50,51]. Since the degradation of lignin is slow and covers a wide temperature range, it overlaps with the other thermal events, thus being present in previous degradation peaks, yet, being unnoticed in the degradation peak analysis, therefore, thermal degradation information on lignin is not explicitly shown in this research. However, observations made by [52] showed that high lignin content was responsible for a high char yield at the end of thermal degradation. At the end of the test (Figure 1), 27% of the initial mass remained, literature reports on charred material content were comparable with the findings of the present study [47,48,49]. Thus, proving the effect of lignin in thermal behavior.

Fique TG and DTG curves are shown in Figure 2. Fique fiber had a first thermal event starting before 100 °C, representing a 5.5% loss of the initial weight linked to the evaporation of water content inside the fiber, as reported by other authors [38,39,53]. The degradation of the main components began at 270 °C and peaked at 297 °C, associated with the decomposition of hemicellulose, representing a 25% weight loss. The onset temperature, degradation peak, and recorded weight loss were similar to fique and other agave fibers [53,54].

As previously stated, the degradation of lignin is a slow process that covers a wide range of temperatures, which, although present, goes unnoticed by the degradation peak analysis; therefore, thermal degradation information on lignin is not explicitly shown in this research. 

Cellulose degradation was registered with a peak at 361 °C, with a weight loss of 30%. The cellulose content and degradation peak were similar for most agave fibers [39,54,55]. An equivalent of 9.4% of the mass persisted at the end of the test.

From the TGA and DTG curves, it is noted that both fibers have different thermal behaviors, which are controlled by their chemical composition. It should be noted that both fibers present relatively low degradation points, however, fique shows slightly higher thermal stability. According to [52], fibers with high cellulose to lignin ratio had an interaction between those two substances that caused incomplete combustion of the fibers, thus acting as flame retardant due to acids being released in the lignin degradation, which could imply that fique may have a greater fire resistance than coir.

#### 4.1.3. Physical and Morphological Parameters

SEM images of the cross-section of the fibers showed that the structure of coir and fique is composed of multiple fibrillar bundles, following the common structure present in vegetable fibers [56]. The cross-section shape of fiber, cell wall thickness, internal lumen size, and shape of individual cells, varied substantially among coir and fique fibers as shown in Figure 3 and Figure 4. Coir displayed a bigger diameter than fique; also, its cross-section showed a circular outline, whereas fique presented highly eccentric elliptical or ribbon-shaped morphology [53,57,58]. Furthermore, different studies show that morphology affects the acoustic performance of fibrous materials. It has been proved that fibers with round cross-sections show greater air permeability, which decreases sound absorption capabilities [59]. Thus, the hypothesis that fique fibers will have superior acoustic performance than coir is raised.

Table 2 shows the measurements of the microstructure of the cross-section of coir and fique fibers obtained from the SEM images using ImageJ measurement tools. Moreover, coir displayed a lower cell wall thickness than fique. Hence, fique could have a better mechanical performance, since a smaller lumen count, and a greater secondary cell-wall thickness is associated with the fiber strength and elastic modulus increase [60]. A *t*-test was performed to determine if the means for cell wall thickness and lumen count were statistically equivalent; the obtained *p*-value < 0.05 and 0.0014; therefore, the null hypothesis is rejected and, thus, the mean cell wall thickness and lumen count in both fibers are different. It should be noted that the number of fiber cells coincides with works of [61].

#### 4.1.4. Density and Diameter Measurements

The average diameter and fineness of coir and fique reported in Table 3, are within the ranges reported in the literature. Average coir diameter showed to be slightly larger than fique, a two-sample *t*-test for equal mean was made, the *p*-value was 0.0064; therefore, the null hypothesis was rejected and, consequently, there is a significative difference between the diameter of both fibers.

Fiber diameter improves the performance of fibrous sound-absorbing materials, increasing flow resistivity as shown in empirical models developed by multiple authors [18,63,64,65]. Therefore, fique fibers would make a superior raw material for acoustic absorption applications.

#### 4.1.5. Mechanical Properties

Table 4 presents the mechanical properties for fique and coir. A *t*-test was made to prove that the means were equal, and the *p*-value obtained was below 0.05, rejecting the null hypothesis; therefore, the average tensile strengths of both fibers are statistically different.

According to literature reports, high cellulose content results in its high tensile strength. Additionally, higher lignin content has been associated with a reduced tensile strength [33,34,35,69,70,71]. Moreover, fibers show high tensile strength if the fibrils are aligned in the fiber axis (0° indicating complete alignment with the fiber axis). Several studies in which fique and coir microfibrillar angles (MFA) were obtained show that fique MFA is considerably smaller than that of coir fibers [37,38,39,44,72,73]. Under those circumstances, fique fibers show mechanical superiority, with a higher cellulose content, a thicker cell wall, and lower lignin content.

Figure 5 shows the stress-strain behavior of the two coir and fique fibers closest to the calculated average. Coir exhibits a ductile behavior, while fique a brittle one; additionally, it can be seen that the elongation of coir is considerably higher than that of fique. The results obtained agreed with those presented in the literature (see Table 4).

### 4.2. Panels Characterization Results

#### 4.2.1. Panels Manufacture

Fique and coir panels were developed by a chemical bonding method; as described in Section 3.2.1, Figure 6 shows the developed materials. Furthermore, Table 5 shows the physical properties of the developed panels.

Considering the results of Table 5, to validate that both means could be assumed as equal, a hypothesis test was made. Thickness and bulk density of the samples were compared using a *t*-test, for which its *p*-value was 0.14 and 0.31, respectively, thus both properties being statistically equivalent with a confidence of 95%, thus validating the fabrication conditions.

#### 4.2.2. Panel Microstructural Observation

Figure 7 and Figure 8 show the microstructure observation of fique and coir panels before and after saturation, respectively. Both materials showed a highly fibrous network, which would be beneficial for acoustic conditioning, with no clear signs of saturated pores between fibers due to water presence. There is evidence of fiber swelling and the presence of a binding agent between fibers due to the applied chemical agent in the images after immersion in which the color changed from transparent to white.

#### 4.2.3. Water Absorption and Thickness Swelling

As fique has a greater content of hydrophilic substances (cellulose and hemicellulose) than coir, consequently fique panels showed a water absorption capacity 1.3 times greater [33,74]. Moreover, according to literature reports, water penetrates the fibers microscopic vascular bundles and remains there by capillary action, causing swelling of the fibers [74,75,76]. The swelling of both panels, shown in Table 6, was compared by a *t*-test; the *P*-value was 0.26, thus accepting that both materials swelling is statistically equivalent. Nevertheless, for maximum water absorption, the null hypothesis was accepted in both case scenarios (*P*-value = 0.25 and 0.45 draining and without draining excess fluid, respectively), indicating that mean water absorption is equivalent for both materials.

As shown in Figure 9, the water absorption for both fibers had a steep increase at the beginning of the experiment, with an absorption over 80% of the total intake in the first 10 min and then plateauing until reaching maximum absorption.

Likewise, the drying rate of both materials had similar behavior as shown in Figure 10, the drying of the samples took over 10,000 min to stabilize. The fique panel ended up with 8% more weight than before the immersion test, whereas coir fibers returned to their original weight. The weights of both fibers took a considerable amount of time to stabilize. It is possible that both panels still held moisture and that excess compensated for water extractables that were partially lost during the immersion test. Nevertheless, it was expected that fique remained more hydrated than coir, due to the major presence of hydrophilic macromolecules [33,34,74].

As panels swell due to water absorption, so do the fibers, also decreasing acoustic absorption capabilities. Therefore, these materials may be suited for conditions with low relative humidity to obtain better performance. Even though fibers have similar behavior, it can be noticed that fique panels retain more moisture, making them inferior under high humidity environments compared to coir.

#### 4.2.4. Mechanical Testing of the Panels

Figure 11 shows the set-up for performing three-point flexural testing on each panel. The load cell of the test reached its extension limit with no rupture in the materials; therefore, the MOR was calculated as previously stated in Section 3.2.4.

Panels flexural stress-deflection curves are shown in Figure 12. Both panels had a ductile behavior, even though none did reach the elastic limit, and once the test was completed, they recovered their original form.

Table 7 shows the modulus of rupture and the *p*-value corresponding to the comparison between the means of such property. Based on the results obtained, there is no statistical difference between both panels modulus of rupture; on the other hand, the material showed a low modulus of rupture.

#### 4.2.5. Sound Absorption Measurement of the Panels

Figure 13 shows the set-up for sound absorption measurements with the Microflown Technologies Impedance Gun. The equipment was screwed to a tripod and placed perpendicular to the reference plane in the center of the sample and 1 cm away from the sample.

Figure 14 shows the average absorption curves for both coir and fique panels. The peak absorption of fique panels is found around 1500 Hz, which represents a quarter wavelength of 2.6 cm if the speed of sound is 344 m/s, followed by an oscillatory behavior common with variations in frequency as it approaches even multipliers of a quarter wavelength. On the other hand, coir panels showed no clear absorption peak, rather a continuous slope.

The results validate the positive effect of small fiber over sample density in sound absorption. The fique panel showed a superior absorption to its counterpart. Additionally, it should be highlighted that coir panels present better absorption than its counterpart from 250 to 800 Hz. However, at low frequencies, sound absorption remains almost unchanged by diameter size; which could be associated with low-frequency absorption isothermal processes favoring density over thickness [77].

Acoustic absorbent materials made from natural fibers can be a promising and environmentally friendly alternative to traditional absorbents. In addition to its acoustic and thermal properties in a material, other factors such as its availability, energy impact, and cost must be considered. Fique and coconut fibers are available in large quantities many times as waste products from other production cycles. Their use can therefore represent a great economic interest for the countries that produce these fibers. The use of local materials contributes to reinforcing their development and use.

They could be used in a wide variety of scenarios. Due to their high porosity and sound absorption, they could be used to reduce the reverberation time in rooms and airborne noise isolation as components of a multilayer system, which could be combined with a panel or perforated board that also provides a rigid, durable, and good-looking surface and improving the acoustic behavior of the structure. Another possibility of using these materials is as part of a floating floor to reduce impact noise as the panels showed that could bend without reaching the elastic limit. The limitations for these materials are mostly high humidity and high temperatures, as they decompose at low temperatures compared to mineral fibers and they retain high amounts of water.

## 5. Conclusions

This research paper deepens the knowledge on fiber properties of two local species in tropical regions, promoting the development of alternative materials that are sustainable by localized production, consumption, and the use of renewable sources and waste fibers as raw materials to a wide audience. Consequently, this research paper developed a characterization and comparison of fique and coir fiber and panels focused on acoustic features. Under this characterization, coir and fique emerge as an alternative, improving the sustainability of the building sector as sound absorption panels with good technical properties and high availability of raw materials, which could stimulate rural economies while providing a better quality of life among city dwellers.

The difference between fibers that are statistically significant translates into the panels sound absorption and the panel water absorption; however, they do not influence the mechanical behavior of the panels.

An apparent association between greater cellulose content and superior tensile strength and fire resistance is raised in this research; also, an association between reduced fiber size and superior tensile strength is shown and, therefore fique fibers present a general superior behavior under the test performed in this research. On the other hand, fique’s greater cellulose and hemicellulose content implies greater water absorption, which could accelerate fungi proliferation and therefore faster degradation than in coir-based materials.

The developed panels showed a highly open porous network, with a low resistance to flexural stress; however, it has a great elongation and did not reach the elastic limit during the test. As a porous network, the material has great water absorption capacity. An advantage of a highly porous network is the potential for sound absorption. Both materials show good absorption capabilities, where fique panels outclass coir in high frequencies. However, the fiber size did not affect the sound absorption at low frequencies, where coir presented better absorption than fique panels.

Future research should be focused on determining which factors in the fabrication method affect the acoustic performance of panels, disposal scenarios, and degradation; study microbial proliferation under target applications, and possible solutions, in addition to testing and improving fire resistance on fibers.

## Figures and Tables

**Figure 1 polymers-13-00101-f001:**
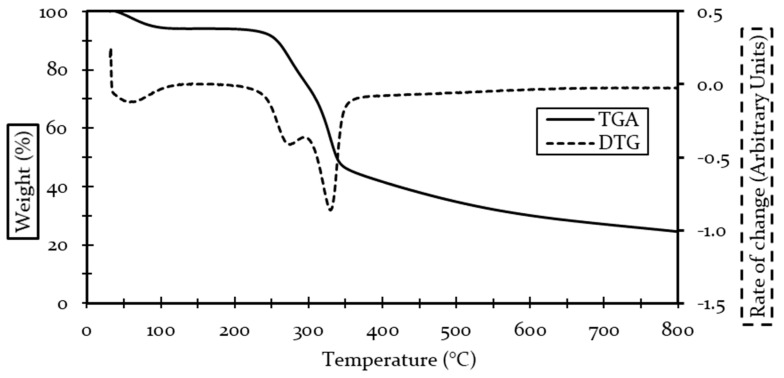
TGA and DTG curves for coir fibers.

**Figure 2 polymers-13-00101-f002:**
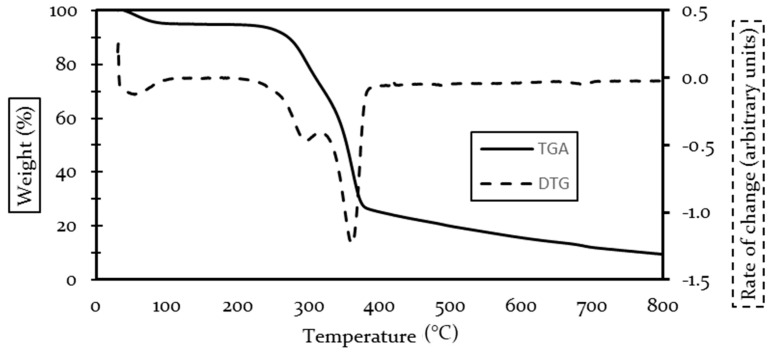
TGA and DTG curves for fique fibers.

**Figure 3 polymers-13-00101-f003:**
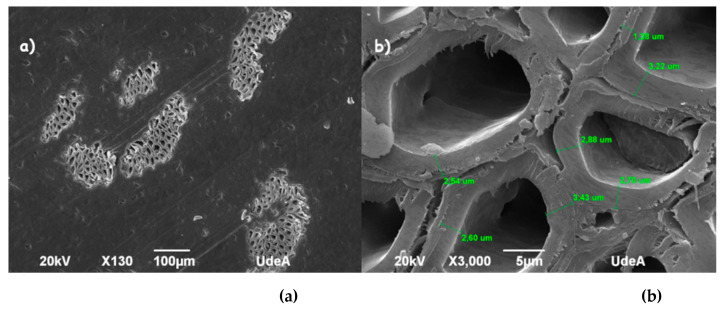
SEM (**a**) micrographs of fique fibers and (**b**) lamella media and lumen of fique fibers.

**Figure 4 polymers-13-00101-f004:**
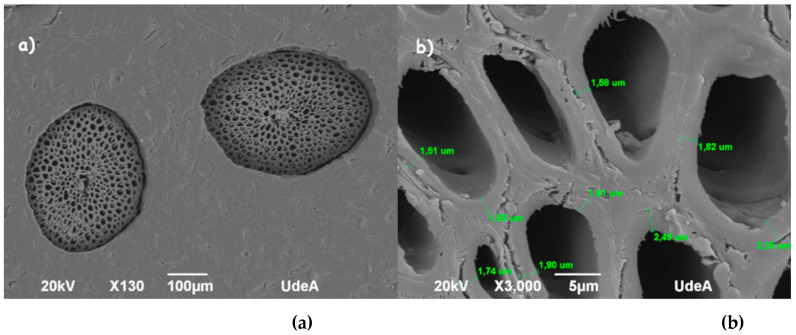
SEM (**a**) micrographs of coir fibers, and (**b**) lamella media and lumen of coir fibers.

**Figure 5 polymers-13-00101-f005:**
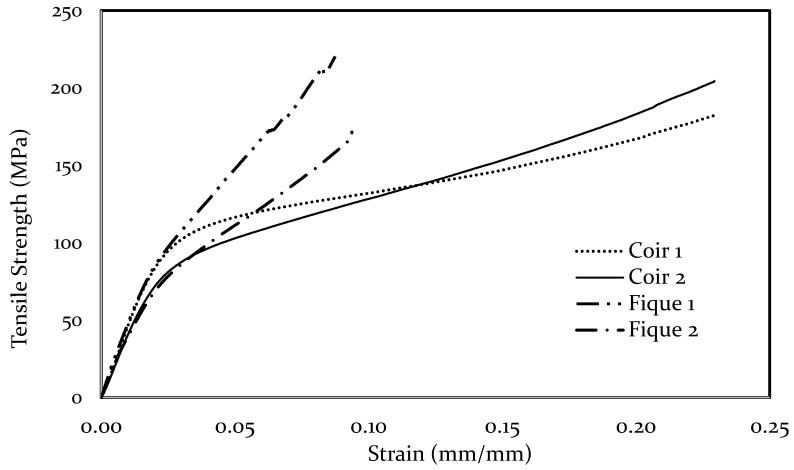
Typical Stress-strain curve of coir and fique.

**Figure 6 polymers-13-00101-f006:**
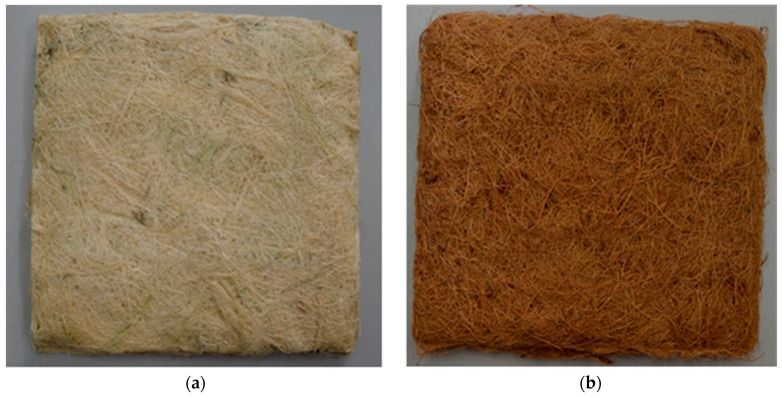
(**a**) Fique and (**b**) Coir panels developed by a chemical bonding method.

**Figure 7 polymers-13-00101-f007:**
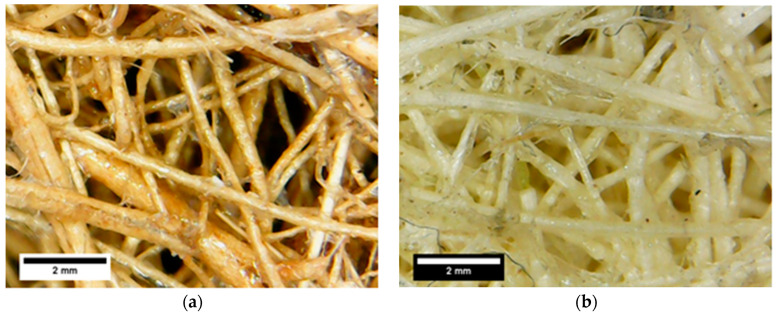
Microstructure of (**a**) dry Coir panel, (**b**) Fique panel.

**Figure 8 polymers-13-00101-f008:**
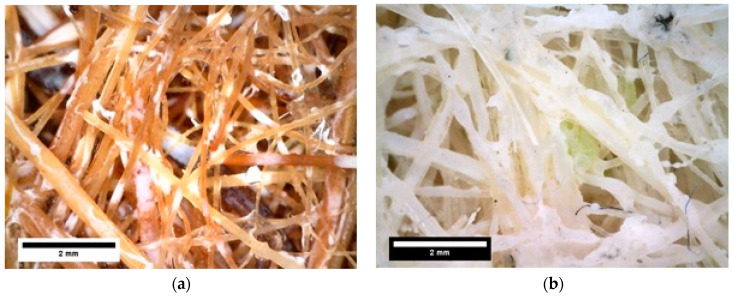
Microstructure after immersion test of (**a**) Coir panel, (**b**) Fique panel.

**Figure 9 polymers-13-00101-f009:**
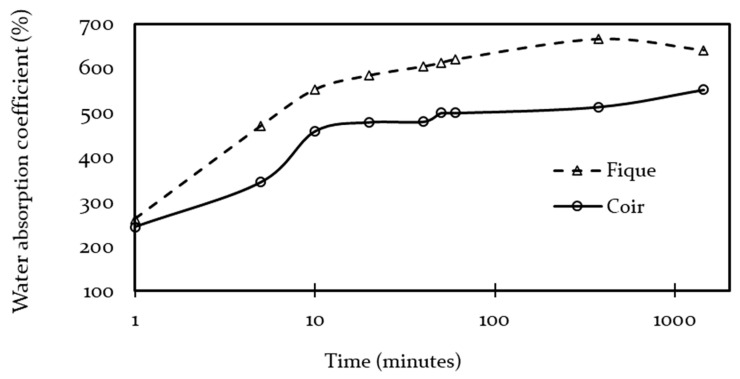
Panel water absorption rate.

**Figure 10 polymers-13-00101-f010:**
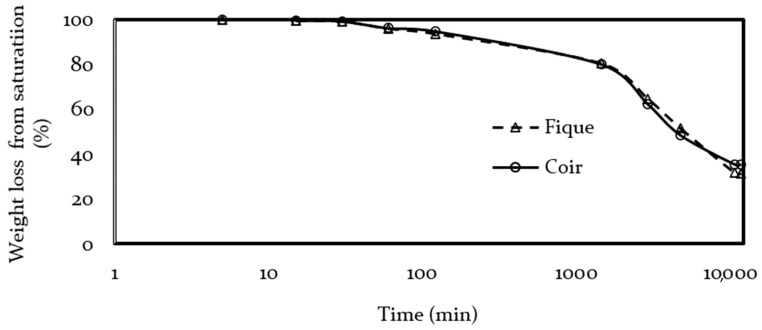
Panels desorption rates.

**Figure 11 polymers-13-00101-f011:**
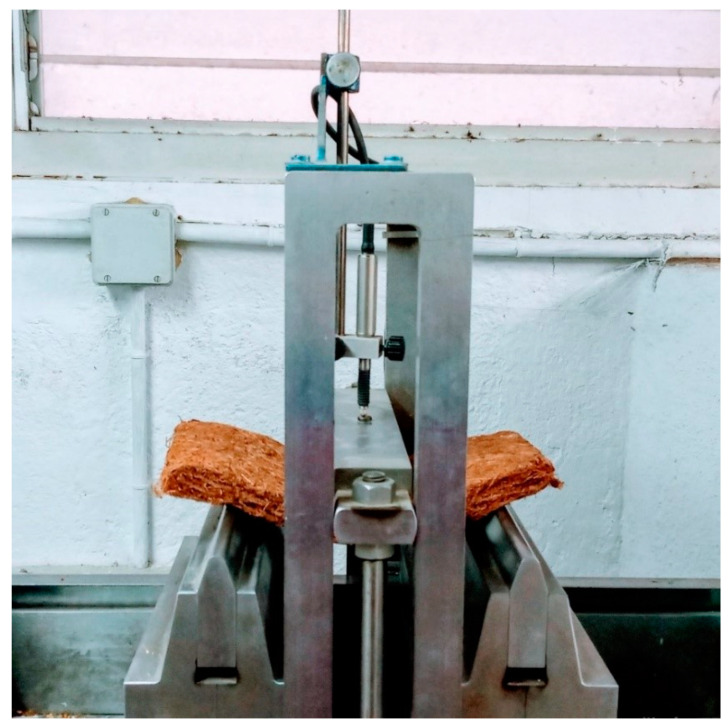
Flexural test set-up.

**Figure 12 polymers-13-00101-f012:**
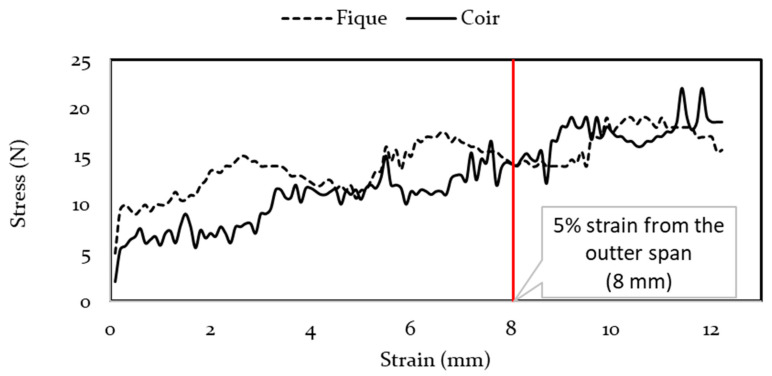
Panels stress-strain curve.

**Figure 13 polymers-13-00101-f013:**
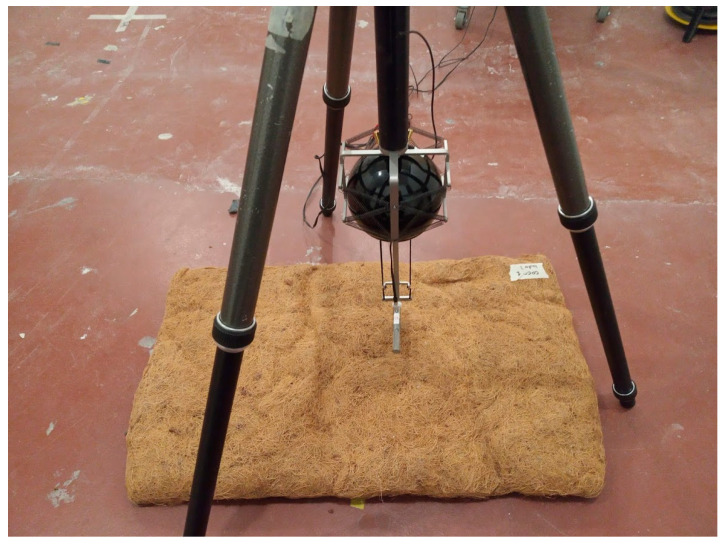
Sound absorption measurement set-up.

**Figure 14 polymers-13-00101-f014:**
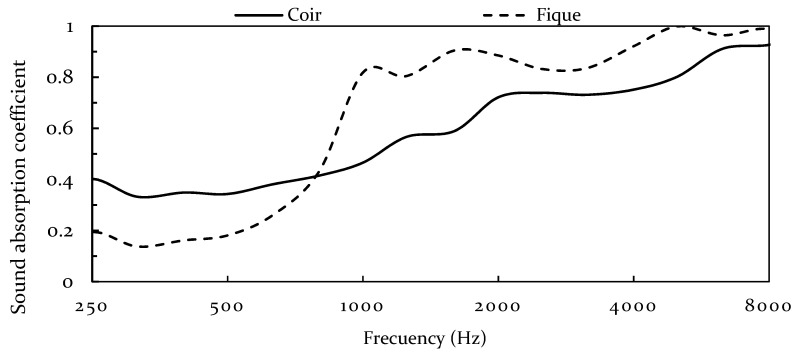
Coir and Fique panels measured sound absorption.

**Table 1 polymers-13-00101-t001:** Chemical composition of coir and fique fibers.

Substance Content	Coir [%]	Fique [%]
Hemicellulose	8.4 ± 2.8	14.6 ± 7.9
Cellulose	46.5 ± 3.5	58.3 ± 17.5
Lignin	33.1 ± 3.6	7.2 ± 3.1
Ash	0.9 ± 0.1	3.7 ± 3.0
Moisture	14.6 ± 0.4	12.8 ± 2.9

**Table 2 polymers-13-00101-t002:** Measurement results of cell wall thickness and the number of fiber cells.

Fiber	Cell Walls Thickness [µm]	Number of Fiber Cells
Coir	1.6 ± 0.5	266.2 ± 41.7
Fique	2.9 ± 1.1	43.7 ± 10.3

**Table 3 polymers-13-00101-t003:** Physical parameters for coir and fique fiber. Present work compared with previous reports.

Fiber	Diameter [µm]	Density [g/cm^3^]	References
Coir	307.5 ± 118.3	0.8	(Present study)
Fique	249.3 ± 111.4	0.6	(Present study)
Coir	252.0 ± 61.3	0.82	[14]
Fique	236.0 ± 63.0	0.72	[62]

**Table 4 polymers-13-00101-t004:** Mechanical properties for coir and fique fibers. Present work compared with the previous report.

Fiber	Tensile Strength [MPa]	Elastic Modulus [GPa]	References
Coir	120.8 ± 67.2	1.8 ± 0.9	(Present study)
Fique	194.8 ± 165.1	4.0 ± 3.3	(Present study)
Coir	162.0 ± 45.0	2.5 ± 0.2	[66,67]
Fique	237.0 ± 51.0	8.0 ± 1.5	[68]

**Table 5 polymers-13-00101-t005:** Panels physical properties.

Fiber	Thickness [cm]	Bulk Density [kg/m^3^]
Fique	2.6 ± 0.5	123.1±33.7
Coir	1.8 ± 0.2	151.7±24.4

**Table 6 polymers-13-00101-t006:** Panels water absorption and swelling.

Fiber	Max Water Absorption (%)	Max Water Absorption Draining Excess Fluid (%)	Moisture Absorbed Bulk Density (kg/m^3^)	Swelling (%)
Fique	653.8 ± 153.5	334.1 ± 38.6	464.5 ± 85,9	13.5 ± 7.7
Coir	560.5 ± 120.4	252.7 ± 87.8	489.9 ± 65.5	6.7 ± 4.2

**Table 7 polymers-13-00101-t007:** Modulus of rupture of panels.

Fiber	MOR (MPa)	*P*-value
Fique	0.020 ± 0.004	0.1
Coir	0.080 ± 0.020	

## Data Availability

Not applicable.

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
