# Peer review of "Evaluation of Colombian Crops Fibrous Byproducts for Potential Applications in Sustainable Building Acoustics"

_polymers, 2020, doi:10.3390/polym13010101_

Round 1

Reviewer 1 Report

The manuscript from the title: "Evaluation of Colombian crops fibrous byproducts for potential applications in sustainable building acoustics", is a very interesting paper.

Namely, the use of lignocellulosic fibers from local species becomes an interesting alternative to the development of such materials. The comparing the physical, morphological, acoustic, and mechanical characteristics of coir (cocos nucifera) and fique (furcraea agavaceae) fibers and panels, the indicates that chemical composition appears to be associated with the general behavior of the fibers and panels, regarding higher tensile strength, thermal degradation behavior, and water absorption. In additional, in most tests, fique had the upper hand, showing superior performance, however, on thermal degradation and water absorption both materials had similar behavior. Even more, an apparent association between greater cellulose content and superior tensile strength and fire resistance is raised, which an association between reduced fiber size and superior tensile strength and flow resistance is shown. Therefore fique fibers present a general superior behavior under the test performed in this research. On the other hand, fique greater cellulose and hemicellulose content implies greater water absorption which could accelerate fungi proliferation and therefore faster degradation than in coir-based materials. Additionally, the according to the sound absorption modeling, the change in diameter due to water absorption does not negatively affect the sound absorption of the developed panels, whereas, the steep increase of density improves such property. This paper deepens the knowledge on fiber properties of two local species in tropical regions, promoting the development of alternative materials that are sustainable by localized production, consumption, and the use of renewable sources and waste fibers as raw materials to a wide audience.

The paper is very easy to read, the experimental procedure is very good and innovative, and the reference list is up to date and more than adequate.

I please the authors only to indicate in results the specific practical fields the applications (industry or other) of the presented innovative materials.

I recommend accept after minor revision (corrections to minor methodological errors and text editing).

Reviewer 2 Report

The submitted work deals with an interesting and important topic. It studies the properties of fiber-based materials, namely, fique leaves and coir, for their potential use in the building industry. The authors study and compare the physical, morphological, chemical, mechanical, and thermal properties of these materials. The authors also examine and compare the properties of panels made of these materials. All these analyses are thorough and well presented.

The authors also try to compare the acoustical properties of panels made of these materials. However, they do not measure the acoustic properties, but they use a very simple model (Delany-Bazley-Miki), which depends on only one parameter – the flow resistivity. Want is more, the authors even do not measure the flow resistivity; they estimate its value employing two parameters: the material bulk density and the fibers’ diameter. The sound absorption properties determined this way are so uncertain that it is hardly possible to make any reliable comparison. For example, there are more accurate phenomenological models (JCA, JCAL, JCAPL), which employ five or more parameters to model the acoustical properties of porous/fibrous materials, whose values strongly depend on the microstructure of these materials.

In practice, the properties of materials used for sound absorption are always measured, for example, in the impedance tube. It is also important to note that the properties of these materials depend not only on the fibers’ diameter but also on their configuration and orientation.

In Fig. 13, the authors show the frequency dependence of the sound absorption coefficient of fique and coir panels backed with a rigid wall.  These curves exhibit local maxima, which correspond to lambda/4, 3*lambda/4, etc., resonances. It means that the thicker the fibrous layer (the panel), the lower the resonance frequencies and the better the low-frequency sound absorption. However, the fique panel is 1.5 times thicker than the coir one! If a reasonable comparison is to be made, the panels must have the same thickness, and their properties must be determined in an experimental way/.

As a result, I do not recommend the publication of this article as it is. The authors should remove all the information about the acoustic properties determined the way described in the manuscript (all the properties except for the acoustic ones are measured, the acoustic properties are modeled), or better, they should team up with an acoustic lab and conduct appropriate measurements.

It seems that Fig. 1 is in the manuscript twice.
